# TIDAL: A Temporal Causal Diffusion Framework for Visualizing Knee Osteoarthritis Treatment Outcomes

## Abstract

Generating realistic patient-specific counterfactual images of treatment outcomes from longitudinal medical imaging is a challenging task, complicated by confounding and selection bias in observational datasets. To address this challenge, we propose TIDAL (Temporal IPW Diffusion Adversarial Learning), a novel longitudinal causal diffusion framework that integrates causal inference techniques directly into diffusion model training. TIDAL utilizes a Stable Diffusion backbone conditioned on patient history and combines two key causal adaptations: (1) Temporal Inverse Propensity Weighting (IPW) that reweights the diffusion loss based on treatment propensity scores; and (2) Domain Adversarial Training that encourages treatment-invariant representations. We demonstrate TIDAL's effectiveness by simulating knee osteoarthritis (OA) progression with longitudinal X-rays from the Osteoarthritis Initiative (OAI). Performance is assessed using image fidelity metrics and observed treatment effects for OA features like Kellgren-Lawrence grade. Our experiments show that TIDAL significantly outperforms baseline approaches, achieving 21.52% reduction in image generation error and 18.43% improvement in observed treatment effects, demonstrating significant improvements for longitudinal medical counterfactual generation.

## 1 Introduction

Visualizing patient-specific future health outcomes under hypothetical interventions holds transformative potential for personalized medicine Huang & Ning (2012); Qian et al. (2021). However, generating faithful counterfactuals from observational medical data faces significant challenges due to confounding bias: factors influencing both treatment assignment and outcomes can lead to spurious correlations and misleading predictions Hernán et al. (2001); Robins et al. (2000).

We focus on knee osteoarthritis (OA), a chronic joint disease affecting 10-37% of people over 60 Sharma (2021); Brophy & Fillingham (2022). Using the Osteoarthritis Initiative (OAI) dataset Nevitt et al. (2006), we propose TIDAL (Temporal IPW Diffusion Adversarial Learning), a framework integrating causal inference techniques into diffusion model training. TIDAL is the first longitudinal causal diffusion framework for patient-specific treatment outcome visualization applied to the OAI dataset.

Traditional causal inference methods focus on tabular data using techniques like Inverse Propensity Weighting (IPW) Robins (1986); Hernán et al. (2001) or representation learning Johansson et al. (2016). Recent deep learning adaptations include sequence models Berrevoets et al. (2021); Melnychuk et al. (2022) and domain adversarial training Melnychuk et al. (2022). While some works have explored counterfactual image generation using diffusion models Sanchez & Tsaftaris (2022); Komanduri et al. (2024); Wang et al. (2024); Yeganeh et al. (2024a), integrating temporal causal inference into longitudinal medical imaging remains underexplored.

**Contributions** In this work, we propose TIDAL, a novel framework that integrates causal inference techniques directly into diffusion model training for longitudinal medical imaging. Our primary contributions are:

1. **TIDAL Framework:** We introduce TIDAL (Temporal IPW Diffusion Adversarial Learning), the first longitudinal causal diffusion framework designed for generating patient-specific, counterfactual medical images over time. We develop two key innovations within TIDAL to mitigate confounding bias:

   (a) **Temporal Inverse Propensity Weighting (IPW):** An RNN-based propensity score model that reweights the diffusion training loss to balance covariate distributions between treatment groups across time.

   (b) **Domain Adversarial Training:** An RNN-based discriminator that encourages treatment-invariant representations in the diffusion model, disentangling image features from treatment selection bias.

   We further combine these two causal mechanisms into an end-to-end learning framework, motivated from a decomposition of the risk function under causal intervention.

2. **Comprehensive Evaluation on Real-World Clinical Data:** We demonstrate TIDAL's effectiveness using the large-scale Osteoarthritis Initiative (OAI) dataset, evaluating both image fidelity and clinical validity through observed treatment effect metrics on clinically relevant features (Kellgren-Lawrence grade, Joint Space Narrowing).

3. **Superior Performance:** TIDAL achieves significant improvements over baseline approaches, with 21.52% reduction in image generation error and 18.43% improvement in X-Ray grade validity, establishing state-of-the-art performance for longitudinal medical counterfactual generation.

## 2 RELATED WORK

### 2.1 COUNTERFACTUAL OUTCOME PREDICTION

Counterfactual outcome prediction has been a crucial task for applications such as personalized medicine and treatment designs Huang & Ning (2012). This task has traditionally been studied under both static and dynamic settings, where the static setting only considers a one-time treatment and the dynamic setting focuses on treatment over time, also known as *longitudinal*. For the static scenario, many existing works have focused on the potential outcome framework Curth & van der Schaar (2021); Johansson et al. (2016); Kuzmanovic et al. (2022); Ma et al. (2024), which focuses on inferring effects such as average treatment effect (ATE) and aims to handle confounding bias and selection bias. A deep learning adaptation of this framework can be found in DiffPO Ma et al. (2024), which addresses the selection bias using a time-agnostic Inverse Propensity Weight (IPW) for tabular data. As for the longitudinal setting, it has been traditionally studied under frameworks such as Marginal Structural Model (MSM) Robins (1986); Robins et al. (2000); Hernán et al. (2001), which rely on linear models. More recently, deep learning based sequence modeling techniques were used, such as recurrent neural networks Qian et al. (2021); Berrevoets et al. (2021), neural ODEs Jiang et al. (2023), and transformers Melnychuk et al. (2022), which also introduce the idea of domain adversarial training to alleviate confounding bias. However, none of these longitudinal methods has been adapted for generating counterfactual images.

### 2.2 COUNTERFACTUAL IMAGE GENERATION IN MEDICAL DOMAIN

Generating counterfactual images differs from standard counterfactual outcome prediction because the predicted target is an image rather than a treatment effect. Due to the added complexity, it is usually more challenging as a learning problem. Existing work in the non-medical domain usually uses causal graphs derived from common sense knowledge Melistas et al. (2024) and employs various generative architectures such as Convolutional Neural Networks for feature extraction Boukhers et al. (2022) or diffusion models for image generation Sanchez & Tsaftaris (2022). Earlier works focusing on causal representation learning Scholkopf et al. (2021) advocated the use of a Structural Causal Model (SCM) in the latent space, which can be even more challenging with the added complexity of image modeling. In the medical domain, existing work focuses more on well-defined treatment and expected outcomes pairs, as well as leveraging features and texts that are rich in medical records Yeganeh et al. (2024a); Wang et al. (2024). Recent work by Glocker and colleagues has made significant advances in high-fidelity counterfactual medical image synthesis Ribeiro et al. (2023), including methods for robust representations via causal image synthesis Pawlowski et al.

(2024) and approaches to mitigate attribute amplification in counterfactual generation Xia et al. (2024). However, these works usually rely purely on probabilistic models such as conditional diffusion models Yeganeh et al. (2024a); Wang et al. (2024) without incorporating causality, leading to confounding and selection bias. Our model combines both the power of a diffusion generative model and the potential outcome framework, hence directly addressing the challenge of counterfactual image generation. To our knowledge, no prior frameworks directly address causal longitudinal image generation for treatment outcome prediction: existing methods are static, and approaches like DiffPO target tabular estimation rather than image synthesis.

## 2.3 Diffusion Models for Causal Inference

Diffusion models Ho et al. (2020); Song et al. (2020) are a class of deep generative models that learn the (often high-dimensional) distribution of the datasets and generate high-quality samples. These models have achieved excellent performance on various computer vision tasks including image synthesis and inpainting Rombach et al. (2021); Dhariwal & Nichol (2021).

**Advantages over VAEs and GANs:** We choose diffusion models for their superior training stability, sample quality, flexible conditioning capabilities, and principled uncertainty quantification compared to GANs and VAEs Ho et al. (2020); Rombach et al. (2021), making them ideal for reliable counterfactual medical image generation.

In the context of causal inference, previous works that focus on counterfactual generation usually rely only on conditional probabilistic inference Yeganeh et al. (2024a); Wang et al. (2024) or injecting a Structural Causal Model into the latent space Komanduri et al. (2024); Sanchez & Tsaftaris (2022). The first approach ignores causality, and the second approach trains a diffusion model from scratch without leveraging existing powerful pre-trained models Rombach et al. (2021). Instead, we fine-tune and correct bias in a pre-trained diffusion model with a causal inference-motivated loss, making our approach both causal and efficient.

## 3 TIDAL: Temporal IPW Diffusion Adversarial Learning

We now present TIDAL, which addresses the key challenges of temporal modeling and confounding bias in longitudinal medical imaging. We present the temporal propensity weighting and adversarial training mechanism separately, then provide the theoretical justification for combining them into an end-to-end learning framework.

### 3.1 Problem Definition

TIDAL aims to generate patient-specific future medical images $X_{t_l}$ at time $t_l$ conditioning on baseline images $X_{t_e}$ from time $t_e$ and treatments $A_{int}$ administered during interval $(t_e, t_l]$, while mitigating confounding and selection bias. TIDAL addresses the counterfactual question: *"what would be the expected outcome, if the patient had received treatment $A_{int}$ during the time interval $(t_e, t_l]$?"* The framework leverages patient longitudinal history $H_{t_e}^{long}$ to train a conditional diffusion model with two novel causal adaptations: Temporal Inverse Propensity Weighting (IPW) and Domain Adversarial Training, which we detail below.

### 3.2 Temporal Conditional Diffusion Model

Diffusion models Ho et al. (2020); Song et al. (2020) progressively add noise to data in a forward process, then learn to reverse this process for generation. The forward process follows $q(x_t \mid x_{t-1}) = \mathcal{N}(x_t; \sqrt{1 - \beta_t} x_{t-1}, \beta_t \mathbf{I})$ with noise schedule $\{\beta_t\}$. The reverse process uses a neural network $\epsilon_\theta$ to predict added noise, trained with objective $\mathcal{L} = \mathbb{E}[\|\epsilon - \epsilon_\theta(x_t, t)\|_2^2]$.

Conditional diffusion models Zhu et al. (2023) accept additional context via classifier labels or text prompts Ramesh et al. (2022). We use embeddings of text prompts extracted from CLIP's text encoder Ramesh et al. (2022) containing treatment and temporal information. Our implementation builds on Stable Diffusion v1.5 which optimizes efficiency by performing diffusion on a lower-dimensional latent space, using a VAE for image compression and reconstruction Rombach

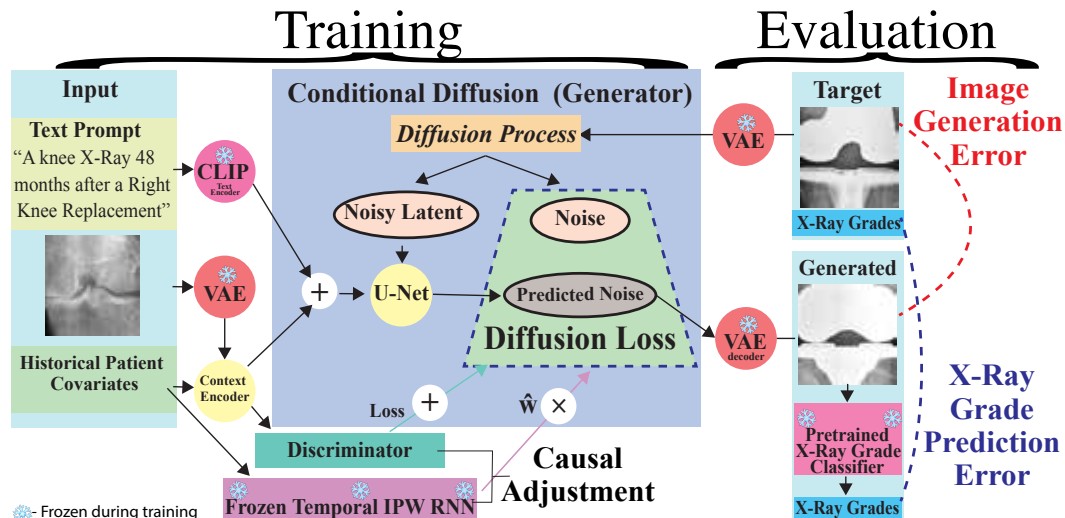

Figure 1: Overview of the TIDAL framework. Inputs (baseline X-ray, patient covariates, text prompt) condition a U-Net for diffusion-based counterfactual generation. TIDAL applies causal adjustment via Temporal IPW RNN weighting and adversarial training with a treatment discriminator. Both components use historical patient covariates to mitigate confounding bias.

et al. (2021). **Core components:** (1) Frozen VAE Kingma & Welling (2022) for latent encoding/decoding, (2) Trainable U-Net Ronneberger et al. (2015) for denoising, (3) Frozen text encoder extracted from CLIP Radford et al. (2021), (4) DDIM scheduler Song et al. (2022).

**U-Net Conditioning Strategy.** To generate a target latent $z_{t_l}$ (corresponding to $X_{t_l}$), the U-Net is conditioned on a combination of textual information and a rich, spatio-temporal context vector:

1. *Textual Prompts ($c_{text}$):* Dynamically generated prompts of the form "A knee X-ray $\Delta t$ months after {treatment list}", where {treatment list} enumerates all treatments within the time interval. These are tokenized and encoded by the CLIP text encoder.

2. *Spatio-Temporal Rich Context ($c_{ctx}$):* This comprehensive conditioning vector is derived by a dedicated Context Encoder RNN ($E_{ctx}$). This encoder processes:

   - The baseline image condition $c_{img}$, which is the output of a linear layer ($f_{img}$) applied to the VAE-encoded latent representation of the baseline X-ray $X_{t_e}$.
   - The patient's longitudinal history $H_{t_e}^{long}$, comprising sequences of historical covariates and treatments up to $t_e$. These sequences are processed by an LSTM Hochreiter & Schmidhuber (1997) within $E_{ctx}$ to capture temporal dependencies, yielding $h_{hist}$.
   - The normalized follow-up duration $\Delta t$ and the knee side $S$, each processed by separate small MLPs to get $h_{\Delta t}$ and $h_S$.

   $E_{ctx}$ concatenates these features, $[h_{hist}; c_{img}; h_{\Delta t}; h_S]$, to form a single vector and is subsequently projected by a linear layer ($f_{proj}$) to match the dimensionality of the text embeddings, resulting in $c_{ctx}$

The final conditioning vector $c_{U\text{-net}}$ fed to the U-Net's cross-attention layers is then the sum of the text embeddings and this projected rich context: $c_{U\text{-net}} = c_{text} + c_{ctx}$.

**Base Diffusion Loss.** The U-Net ($\epsilon_\theta$) is trained to predict the noise $\epsilon$ that was added to the target latent $z_{t_l}$ to produce a noisy latent $z_t$ at timestep $t$. The fundamental training objective for the diffusion process is the Mean Squared Error (MSE) loss:

$$\mathcal{L}_{\text{diffusion}}(\theta) = \mathbb{E}_{z_{t_l}, \epsilon, t, c_{U\text{-net}}} \| \epsilon - \epsilon_\theta(z_t, t, c_{U\text{-net}}) \|_2^2, \tag{1}$$

where we use $\theta$ to denote the trainable parameters for the diffusion generative model.

### 3.3 Inverse Propensity Weighted Diffusion Model

**Standard IPW Background:** Inverse Propensity Weighting (IPW) is a causal inference technique that addresses selection bias by reweighting samples to create a pseudo-randomized population. The propensity score $\pi(x) = P(A = 1 | X = x)$ represents the probability of receiving treatment given covariates. Weighting each sample by $\frac{1}{\pi(x)}$ for treated units and $\frac{1}{(1 - \pi(x))}$ for control units, IPW balances the covariate distributions between treatment groups, simulating a randomized experiment.

**Temporal IPW Extension:** Unlike DiffPO Ma et al. (2024) which uses static covariates, our temporal IPW employs LSTM-based sequence modeling for evolving treatment propensities based on patient history $H_{t_e}^{\text{long}}$. Key differences include: (1) sequential vs. static modeling, (2) interval vs. point treatment prediction, and (3) temporal context integration. Detailed comparisons are provided in Appendix D.

We develop a propensity score model $g_{\phi_p}$ to estimate the probability of receiving a specific set of treatments $\mathbf{A}_{\text{int}}$ during time interval $(t_e, t_l]$, conditioned on the patient's longitudinal history $H_{t_e}^{\text{long}}$, the knee side $S$ (binary indicator for left/right), and the interval duration $\Delta t$. Here $H_{t_e}^{\text{long}}$ is a sequence of patient's historical covariates, time-ordered sequences of tabular features up to $t_e$, including radiographic grades, clinical information, and demographics; $\mathbf{A}_{\text{int}}$ is a sequence of historical treatments, time-ordered multi-hot binary vectors of treatments received up to $t_e$, and $\Delta t$ is normalized (z-score standardized using training set mean and standard deviation) duration $t_l - t_e$.

**Architecture:** The propensity model uses an LSTM (details in Appendix D) to process historical data and output treatment probabilities:

$$\hat{\pi}_k = g_{\phi_p, k}(H_{t_e}^{\text{long}}, \Delta t, S) \approx P(A_{\text{int},k} = 1 \mid H_{t_e}^{\text{long}}, \Delta t, S) \tag{2}$$

**Weighted Diffusion Loss**: After $g_{\phi_p}$ is trained, its weights are frozen. For each sample $i$ in the diffusion model training batch with observed history $H_{i,t_e}^{\text{long}}$, interval duration $\Delta t_i$, side $S_i$, and multi-label interval treatment vector $\mathbf{a}_{i,\text{int}}$, the model $g_{\phi_p}$ provides the estimated marginal probabilities $\hat{\pi}_{i,k}$ for each of the $K$ treatments. Assuming conditional independence of treatment assignments within the interval given the conditioning variables, the joint probability of observing $\mathbf{a}_{i,\text{int}}$ is:

$$\hat{P}(\mathbf{A}_{\text{int}} = \mathbf{a}_{i,\text{int}} \mid H_{i,t_e}^{\text{long}}, \Delta t_i, S_i) = \prod_{k=1}^{K} \left( \hat{\pi}_{i,k}^{a_{i,\text{int},k}} \times (1 - \hat{\pi}_{i,k})^{(1 - a_{i,\text{int},k})} \right) \tag{3}$$

The IPW weight $w_i$ for sample $i$ is the inverse of this joint probability.

$$w_i = \frac{1}{\hat{P}(\mathbf{A}_{\text{int}} = \mathbf{a}_{i,\text{int}} \mid H_{i,t_e}^{\text{long}}, \Delta t_i, S_i)} \tag{4}$$

This weight $w_i$ is then used to modulate the contribution of each sample to the diffusion model's training loss. The IPW-adjusted diffusion loss $\mathcal{L}_{\text{IPW-Diffusion}}$ is calculated as a weighted per-sample diffusion losses:

$$\mathcal{L}_{\text{IPW-Diffusion}} = \sum_{i=1}^{N} w_i \cdot \ell_{\text{diffusion},i}, \tag{5}$$

where $\ell_{\text{diffusion},i}$ is the per-sample diffusion loss in Equation 1.

### 3.4 Domain Adversarial Training

IPW training can introduce unstable training due to some treatment probability values close to zero, which results in exploding IPW weights. To address this issue, domain adversarial methods can be used as an alternative approach to correct confounding and selection bias Lv et al. (2022); Tzeng et al. (2017); Melnychuk et al. (2022). We adapt this method to our longitudinal diffusion model framework by training a treatment discriminator network concurrently with the image generator. The objective is to encourage the generator to learn representations of the baseline patient state ($c_{\text{ctx}}$) that are invariant to the actual treatment $\mathbf{A}_{\text{int}}$ received during the subsequent interval, conditioned on the patient's prior history $H_{t_e}^{\text{long}}$. The adversarial training procedure consists of two network components:

**Diffusion Image Generator (G):** The core conditional diffusion model (detailed in Section 3.2, with trainable parameters $\theta$) is responsible for generating realistic future X-ray images $X_{t_l}$ and the rich context vector $c_{\text{ctx}}$.

**Treatment Discriminator (D):** An auxiliary MLP (parameterized by $\phi$) designed to predict the set of interval treatments $\mathbf{A}_{\text{int}}$ using the generator's context vector $c_{\text{ctx}}$ as input. It outputs $K$ logits, one for each potential treatment.

The training process is conducted adversarially between $D$ and $G$:

- **Discriminator D** aims to accurately predict $\mathbf{A}_{\text{int}}$ from $c_{\text{ctx}}$. For multi-label treatments, it uses $\mathcal{L}_D$, the sum of Binary Cross-Entropy with Logits over the $K$ treatments:

$$\mathcal{L}_D(\phi) = \mathbb{E}_{c_{\text{ctx}}, \mathbf{A}_{\text{int}}} \left[ \sum_{k=1}^{K} \text{BCEWithLogitsLoss}(D(c_{\text{ctx}}; \phi)_k, A_{\text{int},k}) \right], \qquad (6)$$

where $D(c_{\text{ctx}}; \phi)_k$ is the $k$-th logit from **D** for $c_{\text{ctx}}$ (from **G** with fixed $\hat{\theta}$), and $A_{\text{int},k}$ is the true $k$-th treatment label.

- **Generator G** aims to: (1) Minimize the standard diffusion loss $\mathcal{L}_{\text{diffusion}}(\theta)$ (Equation 1) for image quality. (2) Fool **D** by making $c_{\text{ctx}}$ uninformative about $\mathbf{A}_{\text{int}}$. This is achieved via an adversarial loss $\mathcal{L}_{\text{adv}}(\theta)$, encouraging the discriminator's output distribution (for $c_{\text{ctx}}(\theta)$ from **G**) to be uniform by minimizing the negative entropy of **D**'s predicted probability distribution with discriminator parameter $\hat{\phi}$ fixed :

$$\mathcal{L}_{\text{adv}}(\theta) = -\mathbb{E}_{c_{\text{ctx}}} \left[ H(D(c_{\text{ctx}}(\theta); \hat{\phi})) \right], \qquad (7)$$

The overall loss for the generator **G** is a weighted sum:

$$\mathcal{L}_G(\theta) = \mathcal{L}_{\text{diffusion}}(\theta) + \lambda_{\text{adv}} \mathcal{L}_{\text{adv}}(\theta) \qquad (8)$$

where $\lambda_{\text{adv}}$ balances generative quality and adversarial regularization.

**Adversarial Training Procedure:** Parameters $\theta$ (generator) and $\phi$ (discriminator) are updated iteratively:

1. Fix $\theta$, update $\phi$ by minimizing $\mathcal{L}_D(\phi)$.
2. Fix $\phi$, update $\theta$ by minimizing $\mathcal{L}_G(\theta)$.

This encourages **G** to learn $c_{\text{ctx}}$ that is predictive of the outcome image (via $\mathcal{L}_{\text{diffusion}}$) but independent of treatment assignment (via $\mathcal{L}_{\text{adv}}$), conditioned on history, thus mitigating confounding bias.

## 3.5 TIDAL: COMBINED IPW AND ADVERSARIAL TRAINING

The full TIDAL framework combines both temporal IPW and adversarial training to leverage their complementary strengths in addressing confounding bias. While IPW directly reweights samples to balance treatment groups, adversarial training encourages treatment-invariant representations. This results in the combined objective:

$$\mathcal{L}_{\text{TIDAL}}(\theta, \phi) = (1 - \lambda_{\text{adv}})\mathcal{L}_{\text{IPW-Diffusion}}(\theta) + \lambda_{\text{adv}}\mathcal{L}_{\text{adv}}(\theta) + \mathcal{L}_D(\phi) \qquad (9)$$

This overall loss function is motivated by decomposing and bounding the true risk $R^*(\theta)$ associated with the situation of biased treatment assignment $P(A \mid X)$ and representation learning associated with the patient history $H$. Below we state the decomposition formally:

**Setting:** Let $H$ denote patient history, $A$ an intervention, and $X$ the outcome. Observational data follow $q(H, A, X) = p(H)\,p(A \mid H)\,p(X \mid H, A)$. A target (interventional) policy $\pi(A \mid H)$ induces the risk

$$\mathcal{R}^*(\theta) = \mathbb{E}_{p(H)\,\pi(A|H)}\big[\ell(X, f_\theta(H, A))\big].$$

The model uses a representation $Z = g_\theta(H)$ and predicts via $f_\theta(Z, A)$. Define importance weights (IPW) $w(H, A) = \frac{\pi(A|H)}{p(A|H)}$ and an estimate $\hat{w}$. Given samples $(H_i, A_i, X_i) \sim q$, the weighted empirical risk is

$$\widehat{\mathcal{R}}_w(\theta) = \frac{1}{n} \sum_{i=1}^n \hat{w}(H_i, A_i)\, \ell\big(X_i, f_\theta(Z_i, A_i)\big), \qquad Z_i = g_\theta(H_i). \tag{10}$$

Following the standard assumptions of potential outcome framework Rubin (2005) and functional regularity in weighted ERM Cortes et al. (2010), we state the following theorem and provide the technical proof in Appendix E.

**Theorem 1** (Interventional Risk Decomposition)**.** *Let the IPW estimation error be $\varepsilon_{\mathrm{IPW}} :=$ $\mathbb{E}_q\big[(w - \hat{w})\,\ell(X, f_\theta(Z, A))\big]$, representation leakage be $C_\ell\, \mathrm{Disc}(A; Z \mid H) := C_\ell\, \mathbb{E}_{p(H)}\big[D_f\big(p(A \mid Z, H)\,\|\,p(A \mid H)\big)\big]$, and finite-sample generalization error be $\varepsilon_{\mathrm{gen}}(n, W_{\max})$. Assume that:*

*(A1) $\ell \in [0, B]$ or $\ell$ is $L$-Lipschitz in its second argument.*

*(A2) The class $(H, A, X) \mapsto \ell(X, f_\theta(g_\theta(H), A))$ has finite weighted complexity.*

*(A3) Positivity holds (i.e., $p(A \mid H) > 0$ whenever $\pi(A \mid H) > 0$) and the estimated weights are stabilized/clipped so that $\hat{w} \leq W_{\max}$.*

*(A4) Fix a conditional divergence $\mathrm{Disc}(A; Z \mid H) = \mathbb{E}p(H)[D(p(A \mid Z, H), |, p(A \mid H))]$ for an $f$-divergence $D$. Let $C\ell > 0$ depend on $\ell$, $f_\theta$, and the divergence-to-IPM inequality constants.*

*Then for any parameter $\theta$,*

$$\big|\mathcal{R}^*(\theta) - \widehat{\mathcal{R}}_w(\theta)\big| \leq \underbrace{\varepsilon_{\mathrm{IPW}}}_{\text{weighting error}} + \underbrace{C_\ell\, \mathrm{Disc}(A; Z \mid H)}_{\text{representation leakage}} + \underbrace{\varepsilon_{\mathrm{gen}}(n, W_{\max})}_{\text{finite-sample generalization}}, \tag{11}$$

Here the first term corresponds to our proposed $\mathcal{L}_{IPW}$ and the second term corresponds to our design of the domain adversarial loss $\mathcal{L}_{adv}$, justifying our proposed combination of the loss function.

**Training Procedure:** In practice, TIDAL alternates between:

1. *Propensity Model Pre-training:* Train temporal IPW model $g_{\phi_p}$ separately
2. *Joint Training:* Alternate between updating discriminator $\phi$ and generator $\theta$ using the combined loss, with IPW weights applied to both diffusion and adversarial components

This unified approach allows TIDAL to benefit from both explicit propensity-based reweighting and implicit treatment-invariant representation learning, resulting in superior causal performance as demonstrated in our experiments.

## 4 EXPERIMENTS

### 4.1 DATASET CREATION

Our study leverages the publicly available Osteoarthritis Initiative (OAI) dataset Nevitt et al. (2006), a multi-center, longitudinal cohort study focused on knee osteoarthritis (OA). We construct a longitudinal dataset of image pairs and associated clinical information tailored for modeling OA progression and treatment effects.

**Longitudinal Pair and Feature Extraction.** The core of our dataset consists of ordered pairs of X-ray images, $(X_{t_e}, X_{t_l})$, representing an earlier and a later scan for a specific knee of a given patient, where $t_e < t_l$. To maximize data utilization and capture diverse progression intervals, we iterate through each patient and consider all possible chronologically ordered pairs of their available X-ray scans. For each valid pair, we extract a comprehensive set of features, see Appendix B.2:
**Data Splitting.** To ensure subject independence between sets, the dataset is split at the patient level. Unique subject IDs are first divided into a training set (80%) and a temporary set (20%) using a fixed

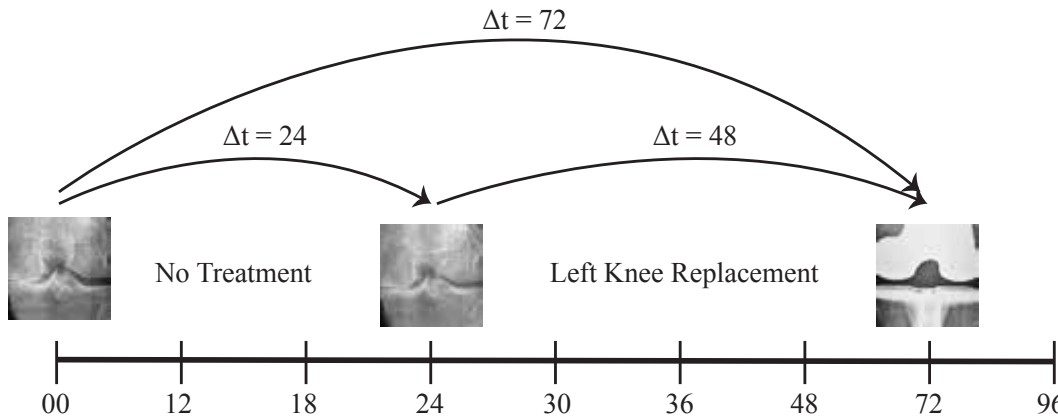

Figure 2: Illustration of longitudinal data pair creation from patient timelines. For each patient, all chronologically ordered pairs of X-ray scans (e.g., month 00 to 24, 00 to 72, 24 to 72) are formed. The interval treatments (e.g., "No Treatment", "Left Knee Replacement") and the duration $\Delta t$ between scans are recorded for each pair.

Table 1: Test performance of treatment outcome modeling with different causal inference adaptations, including percentage decrease in error compared to baseline. Performance on both latent noise representation (Predicted Noise MSE) and image generation (Generated Image MSE and SSIM) are reported.

| Model | Predicted Noise MSE | Generated Image MSE | SSIM |
|---|---|---|---|
| | Value (% vs Base) | Value (% vs Base) | Value (% vs Base) |
| Conditional Diffusion (Baseline) | 0.1361 - | 0.0079 - | 0.77 - |
| IPW Training | 0.1359 (0.15% ↓) | 0.0075 (5.06% ↓) | 0.80 (3.90% ↑) |
| Adversarial Training | 0.1301 (4.41% ↓) | 0.0067 (15.19% ↓) | 0.81 (5.19% ↑) |
| TIDAL | **0.1294 (4.92% ↓)** | **0.0062 (21.52% ↓)** | **0.83 (7.79% ↑)** |

random seed for reproducibility. The temporary set is then further split equally into validation (10% of total subjects) and test sets (10% of total subjects). During training, model state is saved based on the lowest predicted noise loss on the validation set. The model is then evaluated on the test set. See Table 3 for statistics on each split.

## 4.2 EVALUATION METRICS

Like other causal diffusion-based approaches Ma et al. (2024); Song et al. (2022); Ho et al. (2020), we evaluate our framework using predicted noise error and generated mean square error. Generated mean square error is the mean squared error (MSE) between the generated and target image over all the pixels. Additionally, we employ Structural Similarity Index Measure (SSIM) to assess perceptual image quality, which is particularly important for medical imaging where structural preservation is crucial.

Table 2: Performance of X-Ray grade prediction, including percentage decrease in observed treatment effect error compared to Baseline.

| Model | KL Grade | | JSN Medial Grade | |
|---|---|---|---|---|
| | Value | (% Decrease vs Baseline) | Value | (% Decrease vs Baseline) |
| Conditional Diffusion (Baseline) | 0.8152 | - | 0.2996 | - |
| IPW Training | 0.7785 | (4.50% ↓) | 0.2754 | (8.08% ↓) |
| Adversarial Training | 0.7712 | (5.40% ↓) | 0.2511 | (16.19% ↓) |
| TIDAL | **0.7689** | **(5.68% ↓)** | **0.2444** | **(18.43% ↓)** |

**MSE Justification:** While longitudinal radiological images may differ due to non-clinical factors (equipment, positioning, artifacts), MSE remains valuable for: (1) assessing technical quality and anatomical consistency, (2) providing fair comparison across methods, (3) complementing clinical X-Ray grade metrics, and (4) measuring denoising performance in the diffusion framework.

We use observed treatment effect error on factual trajectories to capture clinically significant differences. This error measures differences in clinical X-Ray grades (KL and JSN Medial Grade Kohn et al. (2016)) before and after treatment. We use a pretrained model to predict clinical variables in source, target, and generated images. We then determine the observed treatment effect using the difference between clinical variables in source and target and the predicted treatment effect using the difference between clinical variables in source and generated. We calculate the error using the absolute difference between the observed and predicted treatment effect.

### 4.3 QUANTITATIVE RESULTS

We evaluated TIDAL with four configurations: Baseline, IPW Training, Adversarial Training, and full TIDAL on the test set. Results in Tables 1 and 2 demonstrate causal inference benefits.

Table 1 shows image fidelity metrics. All causal methods improve over baseline. TIDAL achieves best performance: 4.92% reduction in predicted noise error, 21.52% reduction in image error, and 7.79% SSIM improvement. Table 2 shows clinical performance via observed treatment effect error for KL and JSN grades. Lower X-Ray grade error indicates better alignment with ground truth treatment effects. TIDAL demonstrates 5.68% reduction in KL Grade error and 18.43% reduction in JSN Medial Grade error.

These improvements validate our theoretical framework: the IPW component successfully rebalances treatment groups, while adversarial training achieves treatment-invariant representations (shown by consistent improvements across both fidelity and causal metrics). TIDAL's synergistic combination demonstrates that addressing confounding through multiple complementary mechanisms is more robust than individual approaches.

## 5 CONCLUSION

In this work, we presented TIDAL (Temporal IPW Diffusion Adversarial Learning), a novel longitudinal causal diffusion framework that generates patient-specific counterfactual medical images while addressing confounding bias inherent in observational datasets. Our results demonstrate that TIDAL, combining temporal IPW and adversarial training, yields significant improvements in both image fidelity and validity of treatment outcomes. Despite these promising results, our work has several limitations. While we used standard image fidelity metrics, we acknowledge their limitations in fully capturing clinically significant changes in longitudinal medical images; future work should explore more clinically relevant evaluation metrics. Another limitation is that our method is described for counterfactual generation but is evaluated on factual outcomes. While synthetic counterfactual medical datasets exist Khanal et al. (2017); Yeganeh et al. (2024b), to our knowledge, none take into account longitudinal patient information, a critical component for medical utility and treatment-decision making.

Our research carries broader impacts regarding the need for informed patient decision-making in osteoarthritis management. As highlighted by studies showing that patients often lack a clear understanding of potential treatment outcomes Brembo et al. (2016); Pacheco-Brousseau et al. (2021), leading to suboptimal choices, tools that improve patient comprehension are vital. By enabling visualization of patient-specific outcomes under different treatment scenarios, our framework has the potential to enhance clinical decision support and facilitate shared decision-making. This visual aid can empower patients, fostering more informed and appropriate treatment pathways. A prospective clinical trial is needed to rigorously assess its clinical utility and impact on patient decision-making.

However, potential negative impacts require consideration. Risks include the generation of misleading images that could lead to incorrect clinical interpretations if not used responsibly. Fairness is crucial, as performance disparities across diverse patient subgroups could exacerbate healthcare disparities. Finally, the potential for misuse, such as generating fraudulent images, highlights the need for robust safeguards.

TIDAL represents a significant step towards leveraging advanced generative models for personalized treatment outcome visualization, with the potential to ultimately improve patient care and decision-making in osteoarthritis and other longitudinal medical conditions.

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

## A   ADDITIONAL EXPERIMENTAL SETUPS AND RESULTS

### A.1   DATASET DETAILS

Table 3: Summary Statistics of Dataset Splits. Treatment occurrences show the number of image pairs where the treatment was recorded in the interval, with the percentage of total pairs for that split in parentheses. "No Treatment" is inferred for pairs where none of the specified treatments occurred.

| Characteristic | Train | Validation | Test |
|---|---|---|---|
| Image Pairs | 51,726 | 6,684 | 6,331 |
| Unique Subjects | 3,604 | 450 | 451 |
| *Treatment Occurrences (Count (%))* | | | |
| L. Arthroscopy | 933 (1.80) | 132 (1.97) | 153 (2.42) |
| R. Arthroscopy | 963 (1.86) | 131 (1.96) | 95 (1.50) |
| L. Meniscectomy | 702 (1.36) | 130 (1.94) | 138 (2.18) |
| R. Meniscectomy | 775 (1.50) | 76 (1.14) | 85 (1.34) |
| L. Hyaluronic Inj. | 815 (1.58) | 105 (1.57) | 111 (1.75) |
| R. Hyaluronic Inj. | 793 (1.53) | 85 (1.27) | 114 (1.80) |
| L. Steroid Inj. | 1902 (3.68) | 253 (3.79) | 247 (3.90) |
| R. Steroid Inj. | 1805 (3.49) | 182 (2.72) | 302 (4.77) |
| L. Knee Replacement | 679 (1.31) | 108 (1.62) | 93 (1.47) |
| R. Knee Replacement | 704 (1.36) | 51 (0.76) | 108 (1.71) |
| L. Hip Replacement | 411 (0.79) | 34 (0.51) | 26 (0.41) |
| R. Hip Replacement | 417 (0.81) | 93 (1.39) | 48 (0.76) |
| No Treatment | 45,312 (87.60) | 5,845 (87.45) | 5,384 (85.04) |

### A.2   COMMON MODEL ARCHITECTURE AND TRAINING SETUP

All TIDAL variants (Baseline, IPW-enhanced, Adversarially-trained, and combined) share a core generative architecture based on conditional latent diffusion, fine-tuned from Stable Diffusion v1-5 Rombach et al. (2022). All models are implemented in PyTorch, utilizing the PyTorch Lightning framework for training and the Hugging Face Diffusers library for diffusion model components. Training is performed using AdamW optimizers with 16-bit Automatic Mixed Precision (AMP). Shared hyperparameters include a learning rate of 1e-5 for the generator components (U-Net and conditioning MLPs) and a batch size of 64 spread across 2 NVIDIA L40S GPUs. All model variants take up 45,000 MB on each of the two GPUs and take 1.5 days to finish 100 training epochs. The LSTMs used in the Temporal IPW model and Context Encoder had 2 layers with a hidden dimension of 128, they both also used a Dense layer of size 8 for the time delta and 4 for the knee side. Adversarial weight was set to 0.4. All experiments are seeded for reproducibility.

### A.3   QUALITATIVE GENERATED IMAGE EVALUATION

These images were generated by TIDAL with domain adversarial training. During inference, the Stable Diffusion backbone utilized a strength of 0.75, guidance scale of 7.5, and 50 inference steps.

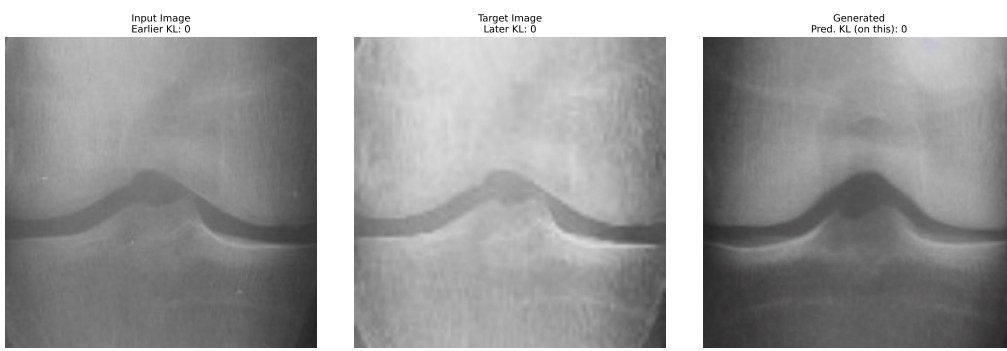

Figure 3: Example X-Ray generated from TIDAL framework

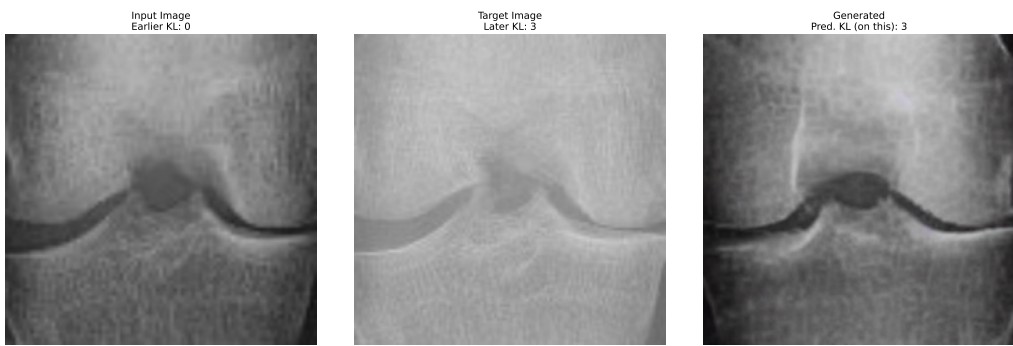

Figure 4: Example X-Ray generated from TIDAL correctly predicting joint space narrowing on right side.

Table 4: Impact of Adversarial Weight ($\lambda_{adv}$) on TIDAL. The reported Validation Loss is the lowest value achieved during training on one validation set for each corresponding adversarial weight.

| Adversarial Weight ($\lambda_{adv}$) | Validation Loss |
| --- | --- |
| 0.8 | 0.1391 |
| 0.6 | 0.1337 |
| 0.5 | 0.1345 |
| 0.4 | **0.1325** |
| 0.2 | 0.1333 |

### A.4 EXAMPLE IMAGES FROM BASELINES

### A.5 ADVERSARIAL WEIGHT ABLATION

## B DIFFUSION PAIR DATASET DETAILS

### B.1 IMAGE PROCESSING AND KNEE LOCALIZATION.

The OAI provides bilateral X-ray images at various timepoints. To focus on individual knee data, we first process these bilateral scans. A YOLOv11-based object detection model Khanam & Hussain (2024), pre-trained on a dedicated knee X-ray dataset for localization Wang et al. (2024); Chen et al. (2019), was employed to detect and crop the left and right knees from each bilateral image see Figure 7. This step ensures that our models receive standardized single-knee views. All cropped images are resized to $224 \times 224$ pixels, converted to tensors scaling pixel values to $[0, 1]$, and then normalized to $[-1, 1]$ (mean 0.5, std 0.5) for input to the diffusion models.

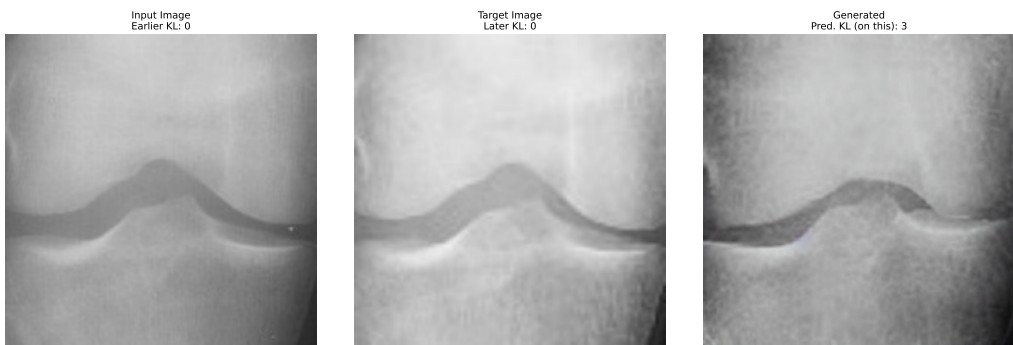

Figure 5: Example X-Ray generated from TIDAL incorrectly predicting joint space narrowing on right side.

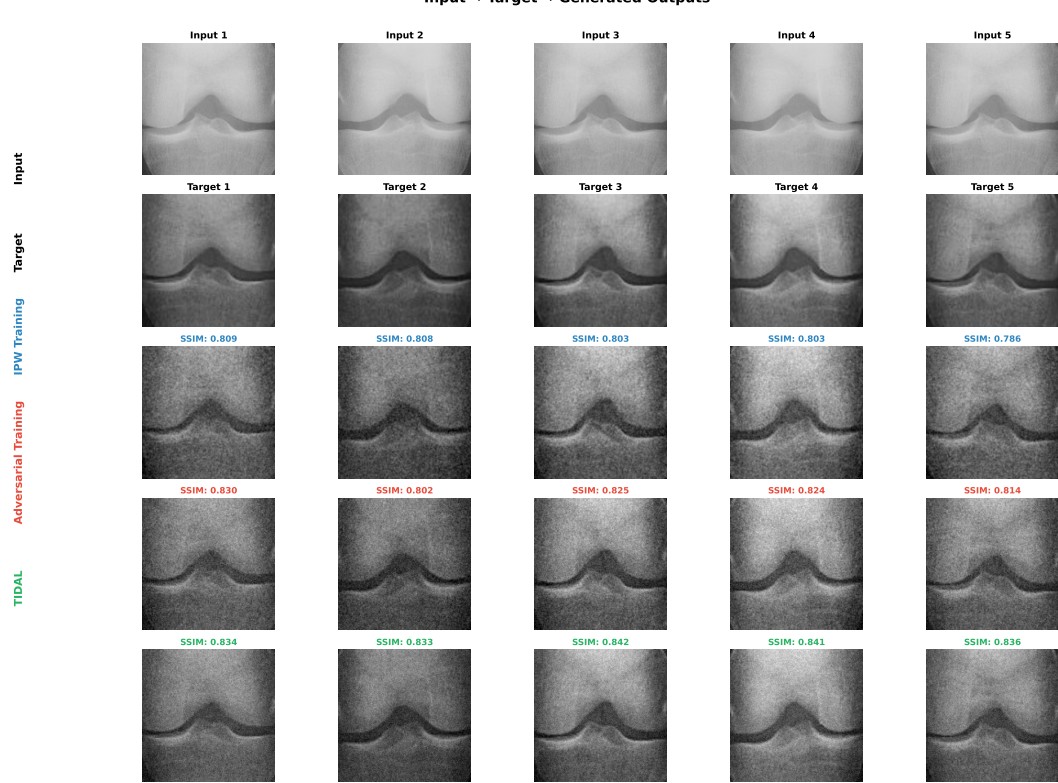

Figure 6: Example X-Rays from 5 randomly chosen inputs from the test set. SSIM from target image and the respective generated image are also reported.

## B.2 EXTRACTED FEATURES.

- **Interval Treatment Information ($A_{int}$):** For a pre-defined list of $K$ treatments (e.g., specific injections, NSAID usage, arthroscopy, knee replacement; $K = 12$ in our setup covering left and right knee treatments such as Arthroscopy, Knee Replacement, Meniscectomy, Steroid Injection, Hip Replacement, and Hyaluronic Injection. This results in a multi-hot vector indicating treatments received during the interval.

- **Radiographic Grades ($H^{tab}$):** Standardized radiological assessments, including Kellgren-Lawrence (KL) grade, and Joint Space Narrowing (JSN) for medial and lateral com-

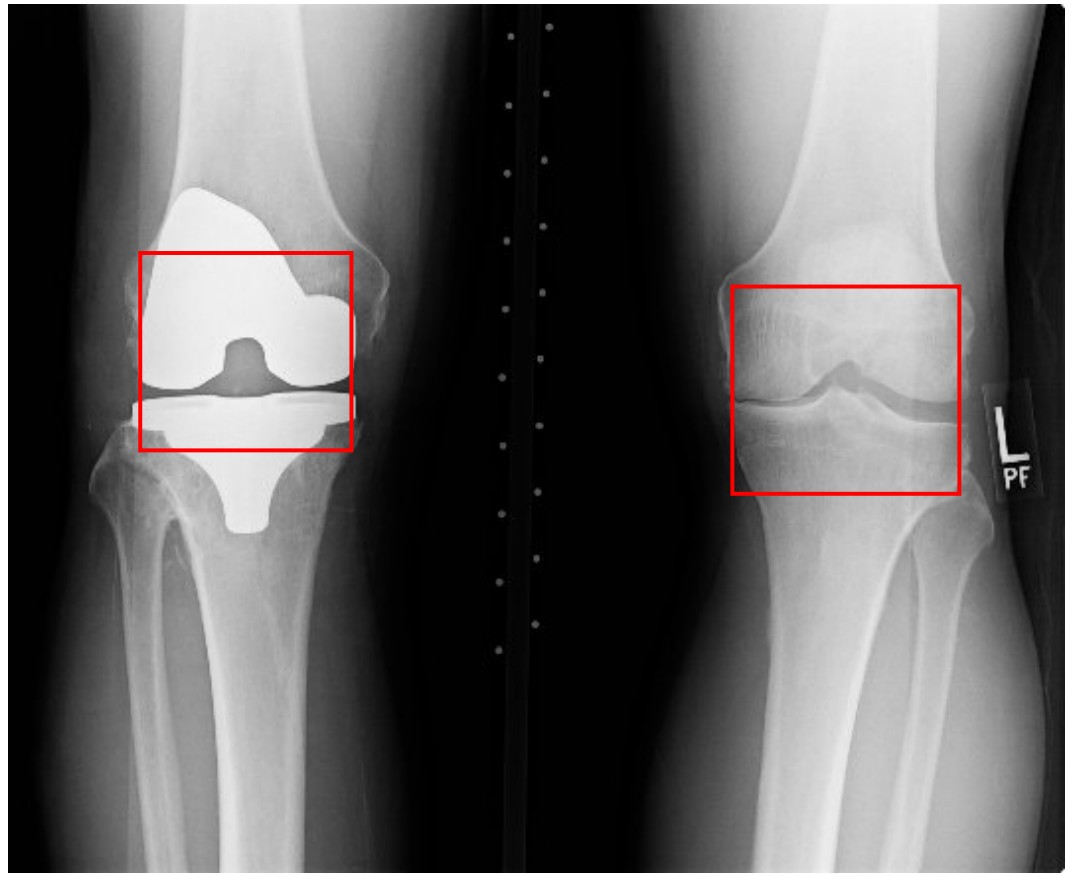

Figure 7: Image showing YOLO detecting bounding boxes for each knee from a Bilateral X-Ray from the OAI dataset.

partments, are extracted for both the left and right knees at both $month\_earlier$ and $month\_later$.

- **Clinical Information:** Time-varying clinical data such as Body Mass Index (BMI) and patient age are recorded.

- **Static Demographics:** Patient-level demographic information like sex, ethnicity, and race are included once per patient.

- **Longitudinal History** ($H_{t_e}^{\text{long}}$)**:** For models utilizing temporal context (IPW and the RNN-based adversarial discriminator), we construct sequences of historical covariates and treatments up to $month\_earlier$.

- **Knee Side** ($S$) and **Follow-up Duration** ($\Delta t = month\_later - month\_earlier$) are also recorded for each pair.

## C  PRETRAINED X-RAY GRADE MODEL DETAILS

To evaluate the causal validity of our generated counterfactual X-ray images, particularly for assessing the generated X-Ray grade prediction error on specific radiographic features, we pre-trained separate classifier models for key osteoarthritis (OA) indicators. We specifically trained models for (KL) Grade and JSN Medial Grade used in our main paper's observed treatment effect evaluations.

## C.1 Dataset and Preprocessing

The feature classifiers were trained using cropped single-knee X-ray images derived from the Osteoarthritis Initiative (OAI) dataset, consistent with the images used for training our main diffusion models. The dataset splits used the same unique patients splits from the Diffusion Model dataset. The specific X-ray grade (e.g., KL Grade ranging from 0-4, JSN Medial from 0-3) served as the target label for each respective model.

Input images were resized to $224 \times 224$ pixels. For training, we applied data augmentation techniques including random horizontal flips, random rotations (up to 10 degrees), color jitter (brightness, contrast, saturation by a factor of 0.2), and random affine transformations (translations up to 10%). All images (for training, validation, and testing) were then converted to tensors and normalized using ImageNet statistics (mean: $[0.485, 0.456, 0.406]$, std: $[0.229, 0.224, 0.225]$).

## C.2 Model Architecture and Training

For each X-ray feature, we fine-tuned a pre-trained EfficientFormerV2-L model Li et al. (2022). The original classifier head of the model was replaced with a new linear layer randomly initialized to output $C$ logits, where $C$ is the number of classes for the specific radiographic feature (e.g., $C = 5$ for KL Grade 0-4, $C = 4$ for JSN Medial Grade 0-3).

The models were trained using a cross-entropy loss function. We employed the AdamW optimizer Loshchilov & Hutter (2019) with an initial learning rate of $1 \times 10^{-5}$. Training was conducted for 30 epochs, and the model state corresponding to the best validation macro-averaged AUC (Area Under the Receiver Operating Characteristic Curve) was saved. The batch size was set to $64$.

## C.3 Performance on Test Set

The performance of the pre-trained classifiers for KL Grade and JSN Medial Grade on the held-out test set is summarized in Table 5. These models are subsequently used in a frozen state to evaluate the observed treatment effect error of the generated counterfactual images from our main diffusion pipelines.

Table 5: Test Set Performance of Pre-trained X-Ray Grade Classifiers.

| Feature | Test Loss | Accuracy | Macro AUC | Num Classes |
|---|---|---|---|---|
| KL Grade | 0.7918 | 0.6724 | 0.8867 | 5 |
| JSN Medial Grade | 1.4385 | 0.8160 | 0.9330 | 4 |

**KL Grade Per-Class Test Accuracy:** {0: 0.858, 1: 0.125, 2: 0.660, 3: 0.843, 4: 0.774}
**KL Grade Class Prevalence (Test Set):** {0: 0.392, 1: 0.175, 2: 0.262, 3: 0.131, 4: 0.039}
**JSN Medial Grade Per-Class Test Accuracy:** {0: 0.902, 1: 0.557, 2: 0.763, 3: 0.822}
**JSN Medial Grade Class Prevalence (Test Set):** {0: 0.669, 1: 0.204, 2: 0.096, 3: 0.029}

# D  Propensity Model Pretraining

We pretrained two distinct propensity models to predict treatment probabilities: a temporal model that incorporates sequential patient history and a non-temporal baseline model. Both models employ RNN architectures but differ significantly in their input representations and temporal modeling capabilities.

## D.1 Temporal IPW vs. DiffPO Comparison

Our temporal IPW addresses fundamental limitations of DiffPO Ma et al. (2024) in longitudinal medical settings:

**Key Differences:** (1) **Sequential vs. Static Modeling**: DiffPO uses time-agnostic propensity models with fixed covariates, while our temporal IPW employs LSTM-based sequence modeling to

capture evolving treatment propensities based on longitudinal patient history $H_{t_e}^{\text{long}}$. (2) **Interval vs. Point Treatment Modeling**: DiffPO predicts single-point treatment assignments, whereas our model estimates probabilities for multi-treatment sets administered during specific time intervals $(t_e, t_l]$, reflecting real-world clinical practice. (3) **Temporal Context Integration**: Unlike DiffPO's static approach, our propensity model incorporates follow-up duration $\Delta t$ and contextual factors (knee side $S$) that influence treatment timing decisions, enabling more accurate propensity estimation in longitudinal settings.

## D.2 MODEL ARCHITECTURES

**Temporal Propensity Model:** The temporal model processes sequential patient histories using an LSTM-based encoder (2 layers, 128 hidden dimensions). We compared LSTM against Transformer architectures, finding that LSTM achieved superior validation performance (AUC: 0.714 vs 0.682 for Transformer). The model takes as input:

- Sequential covariate vectors (medical history over time)
- Sequential treatment vectors (previous treatments)
- Temporal features including normalized time intervals ($\Delta t$) between observations
- Side information (left/right knee distinction)

The LSTM processes concatenated sequence features, followed by specialized MLPs for temporal ($\Delta t$) and side features. The final prediction head combines the sequence encoding with processed features to output treatment probabilities for 13 classes.

**Detailed Architecture:** The LSTM-based propensity model employs the following detailed architecture: The final hidden state from the LSTM $h_{\text{hist}}$ summarizes the patient's entire history. Features for $\Delta t$ and $S$ are processed by separate small MLPs to yield $h_{\Delta t}$ and $h_S$. The concatenated representation $[h_{\text{hist}}; h_{\Delta t}; h_S]$ is passed through a final feed-forward network with sigmoid activation to output the $K$-dimensional probability vector $\hat{\boldsymbol{\pi}} = (\hat{\pi}_1, \ldots, \hat{\pi}_K)$.

**Training Details:** The propensity model $g_{\phi_p}$ is pre-trained separately by minimizing binary cross-entropy loss between predictions $\hat{\boldsymbol{\pi}}$ and true multi-hot interval treatment labels $\mathbf{A}_{\text{int}}$. To address class imbalance, the loss uses positive class weights derived from inverse treatment frequencies in the training data.

**Non-temporal Propensity Model:** The baseline model uses a simpler fusion approach, combining image features from an EfficientFormer backbone with tabular features (X-ray grades, clinical information, and demographics). This model lacks temporal sequence processing and instead operates on static feature representations at individual time points.

## D.3 TRAINING PERFORMANCE COMPARISON

The temporal model demonstrated superior performance across all key metrics:

**Temporal Model Results:**

- Final validation AUC: 0.714
- Final validation accuracy: 68.8%
- Macro recall: 94.1%
- Training converged in 40 epochs with early stopping

**Non-temporal Model Results:**

- Final validation AUC: 0.706
- Final validation accuracy: 62.6%
- Macro recall: 65.4%
- Training completed 50 full epochs

## D.4 KEY FINDINGS

The temporal model's superior performance can be attributed to several factors:

1. **Sequential Information Utilization:** The temporal model leverages the full patient history sequence, capturing temporal dependencies and treatment progression patterns that the static model cannot access.

2. **Temporal Feature Engineering:** The explicit modeling of time intervals ($\Delta t$) between observations, with normalization (mean=35.17, std=22.99), allows the model to understand the temporal spacing of medical events.

3. **Enhanced Recall Performance:** The temporal model achieved significantly higher macro-weighted recall (94.1% vs 65.4%), indicating better identification of patients who actually received treatments.

4. **Class Imbalance Handling:** Both models employed positive weight rebalancing to address the severe class imbalance (90.7% "No Treatment" cases in temporal model), but the temporal model's sequential processing provided better discrimination.

The temporal model's architecture effectively captures the dynamic nature of treatment decisions in longitudinal healthcare data, demonstrating the importance of sequential modeling for propensity score estimation in medical applications.

## E PROOF OF THEOREM 1

Here we restate Theorem 1 in more details and provide a proof sketch.

**Setting.** Let $H$ denote patient history, $A$ an intervention, and $X$ the outcome. Observational data follow $q(H, A, X) = p(H) \, p(A \mid H) \, p(X \mid H, A)$. A target (interventional) policy $\pi(A \mid H)$ induces the risk

$$\mathcal{R}^*(\theta) = \mathbb{E}_{p(H) \, \pi(A|H)}\big[\ell\big(X, f_\theta(H, A)\big)\big].$$

The model uses a representation $Z = g_\theta(H)$ and predicts via $f_\theta(Z, A)$. Define importance weights $w(H, A) = \frac{\pi(A|H)}{p(A|H)}$ and an estimate $\hat{w}$. Given samples $(H_i, A_i, X_i) \sim q$, the weighted empirical risk is

$$\widehat{\mathcal{R}}_w(\theta) = \frac{1}{n} \sum_{i=1}^n \hat{w}(H_i, A_i) \, \ell\big(X_i, f_\theta(Z_i, A_i)\big), \qquad Z_i = g_\theta(H_i).$$

*Theorem 1*: Let the IPW estimation error be $\varepsilon_{\text{IPW}} := \mathbb{E}_q\big[(w - \hat{w}) \, \ell(X, f_\theta(Z, A))\big]$, representation leakage be $C_\ell \, \text{Disc}(A; Z \mid H) := C_\ell \, \mathbb{E}_{p(H)}\big[D_f\big(p(A \mid Z, H) \,\|\, p(A \mid H)\big)\big]$, and finite-sample generalization error be $\varepsilon_{\text{gen}}(n, W_{\max})$. Assume that:

(A1) $\ell \in [0, B]$ or $\ell$ is $L$-Lipschitz in its second argument.

(A2) The class $(H, A, X) \mapsto \ell(X, f_\theta(g_\theta(H), A))$ has finite weighted complexity.

(A3) Positivity holds (i.e., $p(A \mid H) > 0$ whenever $\pi(A \mid H) > 0$) and the estimated weights are stabilized/clipped so that $\hat{w} \leq W_{\max}$.

(A4) Fix a conditional divergence $\text{Disc}(A; Z \mid H) = \mathbb{E}p(H)[D(p(A \mid Z, H), |, p(A \mid H))]$ for an $f$-divergence $D$. Let $C\ell > 0$ depend on $\ell$, $f_\theta$, and the divergence-to-IPM inequality constants.

Then for any parameter $\theta$,

$$\big|\mathcal{R}^*(\theta) - \widehat{\mathcal{R}}_w(\theta)\big| \;\leq\; \underbrace{\varepsilon_{\text{IPW}}}_{\text{weighting error}} \;+\; \underbrace{C_\ell \, \text{Disc}(A; Z \mid H)}_{\text{representation leakage}} \;+\; \underbrace{\varepsilon_{\text{gen}}(n, W_{\max})}_{\text{finite-sample generalization}} \;,$$

*Proof sketch.* Introduce two add–subtract steps and apply the triangle inequality:

$$\left|\mathcal{R}^* - \widehat{\mathcal{R}}_w\right| = \left|\underbrace{\mathbb{E}_q\big[w\,\ell(X, f(H, A))\big]}_{\text{target}} - \underbrace{\tfrac{1}{n}\sum_i \hat{w}_i\,\ell(X_i, f(Z_i, A_i))}_{\text{empirical}}\right|$$

$$\leq \underbrace{\left|\mathbb{E}_q[w\,\ell(X, f(H, A))] - \mathbb{E}_q[w\,\ell(X, f(Z, A))]\right|}_{\text{(Rep)}}$$

$$+ \underbrace{\left|\mathbb{E}_q[w\,\ell(X, f(Z, A))] - \mathbb{E}_q[\hat{w}\,\ell(X, f(Z, A))]\right|}_{\text{(A)}}$$

$$+ \underbrace{\left|\mathbb{E}_q[\hat{w}\,\ell(X, f(Z, A))] - \tfrac{1}{n}\sum_i \hat{w}_i\,\ell(X_i, f(Z_i, A_i))\right|}_{\text{(B)}}.$$

Term (A) yields $\varepsilon_{\text{IPW}}$ by bounded-loss or Cauchy–Schwarz arguments. Term (B) is a weighted ERM concentration term with rate $\tilde{O}(W_{\max}\,\mathfrak{C}/\sqrt{n})$. For (Rep), replacing $H$ by $Z = g(H)$ affects risk only through assignment information. Using Lipschitzness of $\ell \circ f_\theta$ and divergence-to-TV/IPM inequalities, we obtain

$$(\text{Rep}) \;\leq\; C_\ell \,\text{IPM}\big(p(H, A, Z), \tilde{p}(H, A, Z)\big) \;\leq\; C_\ell \,\text{Disc}(A; Z \mid H),$$

where $\tilde{p}$ enforces $p(A \mid Z, H) = p(A \mid H)$. An adversary trained to predict $A$ from $(Z, H)$ provides a variational surrogate that upper-bounds $\text{Disc}(A; Z \mid H)$. Combining the three bounds yields the claim. $\qquad\square$

**Corollary 1** (Justification of the combined objective)**.** *Minimizing the IPW diffusion loss $\widehat{\mathcal{R}}_w(\theta)$ primarily controls $\varepsilon_{\text{IPW}}$, while adding an adversarial invariance penalty (via a discriminator on $(Z, H)$ for predicting $A$) controls $\text{Disc}(A; Z \mid H)$. Regularization/early stopping controls $\varepsilon_{\text{gen}}$. Hence the composite objective*

$$\min_\theta \;\widehat{\mathcal{R}}_w(\theta) \;+\; \lambda_{\text{adv}} \cdot \mathcal{L}_{\text{adv}}(\theta)$$

*is directly aligned with the bound in Theorem 1.*

*Remark* 1 (Space-constrained statement). Under (A1)–(A4), for any $\theta$, $\;|\mathcal{R}^*(\theta) - \widehat{\mathcal{R}}_w(\theta)| \leq \varepsilon_{\text{IPW}} + C_\ell \,\text{Disc}(A; Z \mid H) + \varepsilon_{\text{gen}}(n, W_{\max})$. This motivates IPW (to reduce $\varepsilon_{\text{IPW}}$) and adversarial invariance (to reduce $\text{Disc}$) jointly.

