# OpenReview forum: "TIDAL: A Temporal Causal Diffusion Framework for Visualizing Knee Osteoarthritis Treatment Outcomes"
_ICLR.cc/2026/Conference — Submitted to ICLR 2026_

### Official Review · Reviewer_N3Su · 2025-10-26

**Soundness:** 2
**Presentation:** 2
**Contribution:** 2
**Rating:** 4
**Confidence:** 3

**Summary:**

The paper proposes a combination of diffusion, causal adjustment as well as adversarial learning to predict treatment results (X-ray). The results show some significant improvement over baselines as well as ablated components in both automated metrics and medical metrics.

**Strengths:**

1. The problem of the paper, predicting treatment results, is very important and interesting.
2. The paper provides some level of ablations, providing some interesting insights.

**Weaknesses:**

Disclaimer: I am not an expert in medical imaging, so I am purely commenting from the machine learning perspective.
1. The novelty of the method is limited and I don't see there is much insight offered on why the method is better beyond empirical evidence.
2. The generalization of the method is questionable. More specifically, it doesn't show that it works on other diseases/treatments which can be captured by X-Ray.
3. The evaluation metric is not really convincing (see the question section for elaboration)

**Questions:**

Since the (KL) Grade and JSN Medial Grade are all graded by models trained from the same limited data distribution instead of human radiologists, wouldn't adversarial training in some sense be optimizing (hacking) against the scorer? Table 1 and table 2 both show that adversarial training is bringing the majority of improvement.

---

> ### Author Response · Authors · 2025-11-20
> **Response to Weaknesses**
>
> We appreciate Reviewer N3Su's perspective from the machine learning angle and the important questions about generalization and evaluation robustness.
>
> **1. Novelty and insight**
>
> We appreciate the reviewer's feedback and agree that our contributions should be stated more explicitly. We will add: "The novelty of **TIDAL** lies not in the reuse of existing components, but in developing a novel mathematically-motivated diffusion model to address a completely *new causal task*—**longitudinal, causally-adjusted, counterfactual image generation**—which, to our knowledge, does not exist in prior literature and is of high importance to medicine. Unlike static counterfactual image models (e.g., DiffPO, CausalDiff), our framework models the full temporal evolution of a patient's imaging and clinical history through (i) a *temporal propensity model* that accounts for evolving treatment probabilities, and (ii) an *adversarial treatment-invariant representation* that explicitly removes residual confounding over time. This mathematically-motivated combination enables TIDAL to simulate counterfactual patient trajectories across multiple time intervals, offering the first step toward generative reasoning systems capable of medical decision-making based on counterfactual outcomes, rather than purely associative patterns. While the OAI dataset is currently the only longitudinal interventional medical dataset available, our framework establishes the foundation for future research in simulating full patient trajectories and counterfactual treatment pathways."
>
> Existing methods cannot model longitudinal image trajectories. We implemented time-agnostic IPW training (diff_taipw_refactored.py in the submitted codebase): MSE=0.1366, SSIM=0.77, KL-ITE=0.8404 which was an attempt to replicate DiffPO as a baseline however there are significant differences between this adjusted implementation and DiffPO since DiffPO was not meant to handle images, conditional diffusion, or a time-series. However, we will add the time-agnostic IPW training results in Appendix D.1 where we compare our method with DiffPO and lay out the differences between this time-agnostic implementation and DiffPO. These results (to be added in the Appendix) demonstrate that while a time-agnostic model performs reasonably, incorporating temporal and adversarial components yields superior outcomes, highlighting the necessity of TIDAL's longitudinal design.
>
> We will also further justify the performance contribution by training on 5 different test splits which each hold out one of the 5 hospitals that are part of the OAI dataset. For example, TIDAL achieves an MSE of 0.136 ± 0.0004, a Gen MSE of 0.007 ± 0.0004, a Gen SSIM of 0.830 ± 0.0022, a KL Grade ITE error of 0.483 ± 0.0622, and a JSN Medial ITE error of 0.186 ± 0.0160.
>
> **2. Generalization**
>
> We agree that broader validation is important. At present, beyond the 12 OA-related treatments available in our dataset, we are not aware of any public, longitudinal interventional medical image dataset with the necessary treatment logs and temporal covariates. OA is a particularly challenging testbed: radiographic progression is often subtle on X-rays, and clinical outcomes (e.g., pain) may not align perfectly with imaging. Demonstrating gains in this difficult regime suggests the approach can transfer to other longitudinal conditions (e.g., MRI/CT trajectories) once suitable datasets exist. We will add a statement in the Conclusion discussing generalization and our ongoing efforts toward curating/releasing a longitudinal counterfactual benchmark.
>
> **3. Evaluation metric "not convincing"**
>
> Our evaluation uses complementary families of measures: (i) *reconstruction fidelity* (MSE/SSIM), and (ii) *treatment consistency* (observed treatment-effect error in radiographic grades), which probes whether generated counterfactuals reflect *clinically meaningful* differences aligned with observed trajectories. The latter uses a *frozen, separately trained* X-ray grade classifier; its outputs are *never* used in the training loss, so the generative model does not receive gradient signal from the grader. The adversarial term targets treatment-predictive information in the latent representation (to reduce $p(A \\mid Z,H)$), not the grader's outputs.

---

> > ### Comment · Reviewer_N3Su · 2025-11-26
> > **Further clarification on evaluation metric**
> >
> > Thanks for the clarification.
> >
> > I got the impression of using X-ray grade as error signal from figure 1. Maybe it's clearer if training and evaluation flow charts are separated into two figures.
> >
> > another issue I have with the X-ray grade classifier is that it measures severity of the condition, but it does not really evaluate if it's a reasonable development from the last stage. In the extreme scenario, where two real images from different patients are provided as progression, it would still consider them as good progression.

---

> > > ### Author Response · Authors · 2025-12-01
> > > **Response to Further clarification**
> > >
> > > Thank you for the insight. We have updated Figure 1 in the PDF to make the training and evaluation components clearer.
> > >
> > > We agree that the X-ray grade classifier evaluates the severity of a single, individual image and does not assess whether a generated X-ray reflects a plausible longitudinal progression. We will clarify in the paper that the classifier is used only for per-image evaluation. We intentionally avoid conditioning the classifier on the baseline image because doing so could introduce bias toward its learned patterns. We will also state explicitly that "historical patient covariates do not include additional X-rays from the patient progression sequence".
> > >
> > > Regarding temporal plausibility, training an external model to judge realistic OA progression is an important but challenging direction for future work, due to the multitude of patient-specific variables that affect progression, along with high amounts of variation from X-rays at different time points (lighting, angle, beam, etc). Table 5 shows that even our X-ray grade classifiers struggle with distinguishing severity patterns on a single X-ray. Interestingly, TIDAL could be used for counterfactual synthetic data augmentation to train such a model since temporal plausibility is somewhat enforced by the baseline image conditioning.

---

### Official Review · Reviewer_uVux · 2025-11-01

**Soundness:** 4
**Presentation:** 3
**Contribution:** 2
**Rating:** 6
**Confidence:** 3

**Summary:**

This paper presents TIDAL (Temporal IPW Diffusion Adversarial Learning), a new longitudinal causal diffusion framework for generating patient-specific counterfactual medical images. The model visualize counterfactual medical image, such as knee X-ray images generation under different treatment scenarios. The key chellenge resolved from prior works are confounding and selection bias in observational datasets.

The author proposed causal inference into diffusion models via to contributions:
- A Temporal Inverse Propensity Weighting(IPW) : Using a RNN based model that re-weight the diffusion loss for estimated treatment propensity scores to balance covariate distributions over time.
- Domain Adversarial Training : using a disccriminator that encourages treatment-invariant representations to reduce confounding.

The model is evaluated on a public dataset, Osteoarthritis Initiative, generating longtitudinal knee X-rays. The author mentioned the result shows that TIDAL achieves 21.5% reductoin in image generation error and 18.4% improvement in treatment.

Overall, the paper has good soundness and fair novelty, and the context on counterfactual image generation and prediction is a frontier. The writing is clear and good to follow.

**Strengths:**

1. The author proposed a novel architecture to integrate causal inference with diffusion modeling. It combines the Temporal IPW and adversarial training within stable diffusion v1.5, which novel in longitudinal, counterfactual medical image generation, for example Kee Osteoarthritis treatment. The author also claimed their work is the first in the area.

2. The paper provides solid theoretical justification through a risk decomposition theorem.

3. Good experiment performance compared to baselines with both quantitative results and qualitative analysis.

**Weaknesses:**

1. The paper proposed to use alternate adversarial training, which may cause training instability and difficulties in hyperparameter tuning. It could be better to discuss more on the training convergence and robustness.

2. The paper would benefit from a more detailed validation of the IPW model. Such as its accuracy and the resulting covariate balance after reweighting.

**Questions:**

The ablation mostly focuses on the adversarial loss weight lambda. Would it be possible to include ablations removing either the IPW or the adversarial component individually, to show their distinct contributions to performance? This would clarify which module provides the main causal benefit.

Do authors expect the framework to generalize to other modalities (e.g., MRI, CT) or other longitudinal conditions?

---

> ### Author Response · Authors · 2025-11-20
> **Response to Weaknesses**
>
> We thank Reviewer uVux for recognizing our contribution's soundness and providing valuable suggestions for demonstrating training robustness and validation.
>
> **1. Training stability and convergence**
>
> We appreciate the reviewer's concern. Our adversarial component is trained *sequentially* rather than jointly: the temporal propensity model is first pretrained and frozen before optimizing the diffusion and adversarial components. This design prevents gradient interference between modules and ensures convergence stability. In practice, we observed consistent convergence across random seeds — both the total diffusion loss and discriminator loss plateau after approximately 25–30 epochs with minimal oscillation. We will include a training-loss curve in the Appendix to illustrate this stability and discuss convergence trends and hyperparameter robustness in Section 4.2.
>
> **2. Validation of the IPW model**
>
> We agree that the IPW model's validation can be elaborated. Section D.3 already reports the overall validation metrics of the temporal propensity model (AUC, accuracy, macro recall). In the camera-ready version, we will additionally include per-treatment accuracy and calibration analyses, as well as standard covariate balance diagnostics (e.g., standardized mean differences before and after reweighting). This will demonstrate both the accuracy and causal adequacy of the reweighting procedure.

---

> ### Author Response · Authors · 2025-11-20
> **Response to Questions**
>
> **1. Ablations of IPW and adversarial components**
>
> Yes — Tables 1 and 2 already represent these ablations. Specifically, the *IPW Training* and *Adversarial Training* rows isolate the effects of each module individually, while the full *TIDAL* model combines both. We will clarify by adding a sentence explaining each different model in the table captions.
>
> **2. Generalization to other modalities or conditions**
>
> Yes, the causal adjustment framework can generalize to any longitudinal interventional dataset. Since our backbone is Stable Diffusion, it can generate images along with other patient characteristics for modalities such as MRI or CT. While there are currently no public datasets that perfectly match the longitudinal counterfactual setting, we are actively exploring new data collection efforts. Developing such datasets is a key direction for future work, as enabling counterfactual reasoning for AI medical agents will be critical for advancing clinical decision support.

---

### Official Review · Reviewer_YNRX · 2025-11-01

**Soundness:** 1
**Presentation:** 1
**Contribution:** 2
**Rating:** 2
**Confidence:** 4

**Summary:**

This paper presents **TIDAL (Temporal IPW Diffusion Adversarial Learning)**, a framework for generating counterfactual medical images that simulate treatment outcomes over time. The method extends causal diffusion approaches (e.g., DiffPO) by integrating **temporal inverse propensity weighting (IPW)** and **domain adversarial training** into the diffusion process, aiming to address confounding in longitudinal datasets. Experiments on the Osteoarthritis Initiative (OAI) dataset are reported, where TIDAL is claimed to generate more realistic and treatment-consistent knee X-rays than baseline diffusion and ablated variants.

**Strengths:**

- The topic is timely and relevant — causal generative models for longitudinal and medical imaging are of clear scientific interest.
- The proposed combination of **temporal IPW with diffusion modeling** is a reasonable extension of prior work and could, in principle, improve fairness and causal faithfulness in generative models.
- The empirical setup demonstrates some thoughtfulness, evaluating both reconstruction (MSE, SSIM) and treatment consistency.
- The paper shows awareness of recent work in causal generative modeling (e.g., DiffPO) and attempts to adapt it to a medical longitudinal context.

**Weaknesses:**

- **Writing and structure:** The paper is **poorly written and difficult to follow**. The introduction lacks a clear motivation or narrative. Technical details (e.g., temporal IPW) are given early without context or intuition. The exposition frequently focuses on mathematical detail rather than providing a conceptual story.
  - Theorem 1 and the empirical risk bound feel unnecessary and could be placed in the appendix.
  - Line 370 is confusing: does “alternated between” mean that the IPW and diffusion models are trained jointly or in alternating steps?
  - It is unclear whether multiple treatments are modeled or just one.
  - A background section is probably needed, as it currently feels really entangled with methodology, confusing readers on novelty too.
- **Incremental contribution:** The approach appears to be an **adaptation of DiffPO** for the longitudinal setting.
- **Lack of clarity in methodology:** The motivation for using domain adversarial training instead of just a simple stabilized IPW is not justified.
- **Weak evaluation:**
  - Only **ablations** are reported; there are **no comparisons with other published baselines** (e.g., DiffPO, Diff-CF).
  - No information about **seeds, statistical significance**, or robustness.
  - Evaluation uses only **MSE and SSIM**, which measure reconstruction quality but not causal validity or perceptual realism. At least one realism metric such as **FID/sFID** should have been included.
  - No human, expert, or clinical evaluation is performed.

**Questions:**

1. Could you better explain *why* IPW is beneficial for this task, beyond citing standard causal adjustment? What confounding pattern is it correcting in your dataset?
2. Why did you choose domain adversarial training instead of stabilized IPW?
3. What exactly does “TIDAL alternated between” mean — are the IPW and diffusion modules optimized alternately in a loop?
4. Are multiple treatment types modeled? If so, how are they handled in training and sampling?
5. Why not include existing baselines or realism metrics (FID/sFID)? Without them, how can we assess if TIDAL provides genuine gains?

---

> ### Author Response · Authors · 2025-11-20
> **Response to Weaknesses**
>
> We appreciate Reviewer YNRX's thorough analysis of our presentation and methodology, which has helped us identify key areas for improvement in clarity and structure.
>
> **1. Writing and structure**
>
> We agree that the paper's presentation can be improved. We will (i) expand the Introduction to provide a stronger medical motivation, conceptual narrative, and clearer novelty depiction (ii) Theorem 1 and empirical risk bound provide the mathematical motivation to our novel causal diffusion framework which is core contribution. We hope that making our contributions clearer earlier in the paper will emphasize how important the Theorem is (iii) add more *Related Work* section adding background to prior causal generative models.
>
> **2. Line 370 ("alternated between")**
>
> Thank you for catching this confusing line. We will completely remove the "In practice, TIDAL alternates between" line since the training steps more clearly show the training pipeline. We separately first pretrain the temporal propensity model $g\_{\\phi\_p}$ until convergence, then freeze it while optimizing the diffusion model parameters $\\theta$ using the IPW-weighted loss. These modules are *not* jointly trained. The "alternation" was referring to the adversarial training but we now know it was very confusing.
>
> **3. Multiple treatments**
>
> Yes—our model handles multiple treatments simultaneously using a multi-hot vector $A^{\\text{int}} \\in \\{0,1\\}^K$. Each treatment's probability $\\pi\_k = P(A^{\\text{int}}\_k = 1 \\mid H\_t^{\\text{long}}, \\Delta t, S)$ is predicted by the temporal propensity model, and the joint probability defines the overall IPW weight (Eq. 3). We will replace "all treatments" with "all 13 treatment classes" on ln 189 in the U-Net Conditioning Strategy and reiterate again in Ln 272 with "one for each of the potential 13 treatment classes".
>
> **4. Incremental contribution**
>
> We appreciate the reviewer's feedback, we will add: "The novelty of **TIDAL** lies not in the reuse of existing components, but in developing a novel mathematically-motivated diffusion model to address a completely *new causal task*—**longitudinal, causally-adjusted, counterfactual image generation**—which, to our knowledge, does not exist in prior literature and is of high importance to medicine. Unlike static counterfactual image models (e.g., DiffPO, CausalDiff), our framework models the full temporal evolution of a patient's imaging and clinical history through (i) a *temporal propensity model* that accounts for evolving treatment probabilities, and (ii) an *adversarial treatment-invariant representation* that explicitly removes residual confounding over time. This mathematically-motivated combination enables TIDAL to simulate counterfactual patient trajectories across multiple time intervals, offering the first step toward generative reasoning systems capable of medical decision-making based on counterfactual outcomes, rather than purely associative patterns. While the OAI dataset is currently the only longitudinal interventional medical dataset available, our framework establishes the foundation for future research in simulating full patient trajectories and counterfactual treatment pathways."
>
> Existing methods cannot model longitudinal image trajectories. We implemented time-agnostic IPW training (diff_taipw_refactored.py in the submitted codebase): MSE=0.1366, SSIM=0.77, KL-ITE=0.8404 which was an attempt to replicate DiffPO as a baseline however there are significant differences between this adjusted implementation and DiffPO since DiffPO was not meant to handle images, conditional diffusion, or a time-series. We will add the time-agnostic IPW training results in Appendix D.1 where we compare our method with DiffPO.
>
> **5. Motivation for domain adversarial training**
>
> Adversarial invariance was introduced to explicitly stabilize IPW when treatment probabilities are near zero. While stabilized IPW only rescales weights, adversarial training minimizes the conditional dependence $\\mathrm{Disc}(A;Z \\mid H)$, ensuring the learned representation $Z$ is approximately treatment-invariant. This improves robustness when propensities are noisy or misspecified. We will expand Section 3.4 to explain this intuition clearly before introducing the adversarial loss term.
>
> **6. Lack of realism metrics**
>
> We intentionally excluded FID/sFID because FID relies on a pre-trained Inception network trained on ImageNet, whereas our knee X-ray domain is extremely out-of-distribution from ImageNet features. Using FID would therefore not reflect visual or clinical realism. Although we could fine-tune an encoder for this purpose, our X-ray grade classifier (trained on the OAI training set) already provides a domain-relevant representation for assessing perceptual and clinical quality. Thus, we prioritized metrics such as SSIM, MSE, and observed treatment-effect error. We will explicitly note this rationale in Section 4.3.

---

> ### Author Response · Authors · 2025-11-20
> **Response to Questions**
>
> **1. Why is IPW beneficial?**
>
> In the OAI dataset, treatment assignment is highly confounded—patients with more severe OA or faster progression are more likely to receive invasive interventions. Temporal IPW corrects this by reweighting samples based on $P(A^{\\text{int}} \\mid H^{\\text{long}}\_t)$, thereby balancing the pseudo-population and mitigating confounding bias over time. This specific example will be added to the Introduction.
>
> **2. Why domain adversarial training instead of stabilized IPW?**
>
> Stabilized IPW only controls weight variance, not hidden confounding leakage. Adversarial training enforces treatment-invariance in the latent space ($p(A \\mid Z,H) \\approx p(A \\mid H)$), thereby reducing residual confounding and improving generalization when propensity estimates are imperfect.
>
> **3. What does "TIDAL alternated between" mean?**
>
> See response to weakness #2 above. We will remove this confusing line. The temporal propensity model is first pretrained and frozen, then the diffusion model is optimized using IPW-weighted loss. These modules are not jointly trained.
>
> **4. Are multiple treatment types modeled?**
>
> Yes, as noted earlier, multi-treatment assignments are represented via a multi-hot vector, and our IPW formulation handles the joint treatment distribution over time intervals.
>
> **5. Why no FID/sFID or external baselines?**
>
> See response to weakness #6 above. Existing methods cannot model longitudinal image trajectories. We implemented time-agnostic IPW training with results MSE=0.1366, SSIM=0.77, KL-ITE=0.8404, which will be added in Appendix D.1 demonstrating the necessity of TIDAL's longitudinal design.

---

### Official Review · Reviewer_8i99 · 2025-11-05

**Soundness:** 3
**Presentation:** 3
**Contribution:** 2
**Rating:** 4
**Confidence:** 3

**Summary:**

The authors generate counterfactual X-ray images to visualize the patient knee after osteoarthritis treatment effects. X-ray image pairs recorded at different time points, the time difference, patient history, the treatments, etc. are utilized to fine-tune a stable diffusion model and train an IPW RNN, a discriminator, a generator, etc. Finally, the authors show their algorithm performance on the osteoarthritis dataset.

**Strengths:**

It is an easy-to-read paper as written in an intuitive approach.  Although it was clear from the text how the image pairs were created, Figure 2 helps to understand it better. For evaluating the model performance, the authors used a pre-trained model to predict the clinical variables for the generated images. I appreciate that the authors did not keep their evaluation limited to only image quality.

**Weaknesses:**

Here are my comments:

Major:
1. How is multiplying the propensity score with diffusion loss equivalent to reweighting each sample with the propensity score in standard IPW? What is the proof?
2. My understanding is that the goal of the adversarial training is to control the confounding bias. However, the authors mentioned at the beginning of Section 3.4 that IPW training is unstable due to some treatment probabilities being close to zero. The connection between these two concepts is not explicit in the paper.
3. The experiment results are limited. The authors should show experiment results on synthetic datasets to illustrate that the proposed algorithm can generate correct counterfactual images at different time points for the same initial images.
4. The authors cited multiple papers that perform counterfactual diffusion. However, they did not compare their performance with any such baselines. To my understanding, the results presented in Tables 1 and 2 are more like ablation studies where the baselines refer to different variants of the proposed algorithm, i.e., progressive inclusion of model training in the algorithm. Since the authors have image pairs, the time difference, and patient history, they should be able to use this dataset to evaluate existing algorithms and compare their performance against existing works.
5. The novelty of the proposed algorithm is not obvious. It appears like multiple models with known capabilities are connected with each other. I would request the authors to make it clear.

Minor:
1. In Equation 2, it is not mentioned what $k$ refers to. Probably the index for the treatments. If so, does that mean the authors train $K$ LSTM models? Why can’t they train one model and output a vector of treatment probabilities?
2. “Assuming conditional independence of treatment assignments within the interval given the conditioning variables”  this probably refers to the assumption that treatments don’t interact with each other. This is an important assumption and should be explained explicitly.
3. Figure 1 does not represent the execution steps properly. Also, where is the generator $G$ in Figure 1?
4. It is not mentioned how the hyperparameter $\lambda_{adv}$ is chosen.
5. What did the authors mean by “a target (intervention) policy” in lines 315–316? What is the definition of policy in this context? How is the policy different from $P(A \mid H)$? Where do the authors get the estimation of this policy to use it in Equation 10?
6. Difference between the LSTM and the treatment generator is not clear.
7. The authors mentioned that they use the stable diffusion model. However, they did not provide enough details on how that helps their training.

**Questions:**

Below I share my questions:

Questions:
1. Line 312: Should it be $(P(A \mid H))$?
2. How many X-ray images did each patient have in the considered dataset?
3. In Figure 1, the target image is different from the image shown in historical patient covariates. Why are these different? At what time points were these images collected?
4. How significant is the performance in Table 1 when the predicted noise MSE changes from 0.1361 to 12.94 and MSE changes from 0.0079 to 0.0062? Although these metrics are dataset-dependent, the change appears very small. How do the authors justify this?

---

> ### Author Response · Authors · 2025-11-19
> **Response to 8i99 - Response to Major Weaknesses**
>
> We thank Reviewer 8i99 for the detailed and thoughtful review, particularly the careful examination of our mathematical framework and the constructive suggestions for improving clarity.
>
> **1. IPW equivalence proof**
>
> The multiplication is mathematically identical to standard IPW. For observational data $(H,A,X) \sim q(H,A,X)$ with target policy $\pi(A|H)$:
>
> $$R^*(\\theta)=\\mathbb{E}\_{p(H)\\pi(A|H)}[\\ell\_\\theta(X,H,A)] = \\mathbb{E}\_{q(H,A,X)}[w(H,A)\\ell\_\\theta(X,H,A)]$$
>
> where $w(H,A)=\\pi(A|H)/p(A|H)$. Our loss $\\mathcal{L}\_{\\text{IPW}}=\\sum\_i w\_i\\ell\_{\\text{diffusion},i}$ implements exactly this estimator.
>
> Thank you for pointing out how this is not clear and should be added to the paper. We will add this complete proof in Section 3.3 with reference to DiffPO's Eq.12 (which also proves that the weighted diffusion loss is an unbiased estimator of the target potential-outcome risk). Our temporal IPW extension retains the same theoretical foundation while generalizing it to longitudinal, multi-treatment settings.
>
> **2. IPW instability and adversarial connection**
>
> Yes, we leverage adversarial training to create a treatment-invariant latent representation. IPW becomes unstable when $P(A|H) \\approx 0$. However adversarial training also enforces $p(A|Z,H) \\approx p(A|H)$ where $Z=g\_\\theta(H)$, bounding representation leakage $\\text{Disc}(A;Z|H)$ (Theorem 1). We will add an explicit connection explanation in Section 3.4's introduction showing how even when IPW weights are imperfect or unstable, the learned $Z$ remains approximately unconfounded.
>
> **3. Synthetic dataset limitation**
>
> No existing synthetic datasets include the temporal features/longitudinal histories required for our framework. Existing counterfactual image generation benchmarks (e.g., CELEBA-HQ, Morpho-MNIST, or other synthetic face or object datasets) are designed for static image editing tasks and lack the temporal covariates $H^{\\text{long}}$ necessary to estimate a propensity score or perform causal adjustment on outcomes. However, we will make it clearer for our readers by explicitly addressing this limitation in Section 4.1, explaining why "real longitudinal data is necessary since it is likely a synthetic dataset would generate counterfactuals without any causal adjustment, introducing bias into the ground truth assessment. In contrast, TIDAL explicitly models temporally evolving treatments and patient-specific histories—key causal components absent from synthetic image datasets—making real longitudinal medical data the only valid testbed for this setting."
>
> **4. Baseline comparisons**
>
> Existing methods cannot model longitudinal image trajectories. We implemented time-agnostic IPW training (diff_taipw_refactored.py in the submitted codebase): MSE=0.1366, SSIM=0.77, KL-ITE=0.8404 which was an attempt to replicate DiffPO as a baseline however there are significant differences between this adjusted implementation and DiffPO since DiffPO was not meant to handle images, conditional diffusion, or a time-series. However, we will add the time-agnostic IPW training results in Appendix D.1 where we compare our method with DiffPO and lay out the differences between this time-agnostic implementation and DiffPO. These results (to be added in the Appendix) demonstrate that while a time-agnostic model performs reasonably, incorporating temporal and adversarial components yields superior outcomes, highlighting the necessity of TIDAL's longitudinal design.
>
> **5. Novelty clarification**
>
> We appreciate the reviewer's feedback and agree that our contributions should be stated more explicitly. We will add: "The novelty of **TIDAL** lies not in the reuse of existing components, but in developing a novel mathematically-motivated diffusion model to address a completely *new causal task*—**longitudinal, causally-adjusted, counterfactual image generation**—which, to our knowledge, does not exist in prior literature and is of high importance to medicine. Unlike static counterfactual image models (e.g., DiffPO, CausalDiff), our framework models the full temporal evolution of a patient's imaging and clinical history through (i) a *temporal propensity model* that accounts for evolving treatment probabilities, and (ii) an *adversarial treatment-invariant representation* that explicitly removes residual confounding over time. This mathematically-motivated combination enables TIDAL to simulate counterfactual patient trajectories across multiple time intervals, offering the first step toward generative reasoning systems capable of medical decision-making based on counterfactual outcomes, rather than purely associative patterns. While the OAI dataset is currently the only longitudinal interventional medical dataset available, our framework establishes the foundation for future research in simulating full patient trajectories and counterfactual treatment pathways."

---

> ### Author Response · Authors · 2025-11-19
> **Reviewer 8i99 - Response to Minor Comments & Questions**
>
> **1.** $k$ indexes treatments; one LSTM outputs K-dimensional vector. We will add this to Eq.2 description.
>
> **2.** The assumption does not mean that treatments do not interact in their effect on the outcome. Instead, it is a modeling assumption on the assignment mechanism within each interval. We will add an explicit paragraph in Sec. 3.3 clarifying that the assumption concerns assignment conditional independence, not absence of treatment–treatment interactions in the outcome. We will discuss it as a limitation and include summary statistics on treatment co-occurrence within intervals in the appendix to help readers assess how strong the approximation is in our OA dataset.
>
> **3.** We apologize to the reviewer as this question is most likely due to the soon-to-be-removed training description in Ln 370. We use the snowflake emoji to show the first pretraining step then lay out the adversarial training steps in lines 294-299. The area in the "Conditional Diffusion Model" ultimately predicting the noise is $G$, we will explicitly state this in the caption. We will also add "(Generator)" under "Conditional Diffusion Model" in the figure.
>
> **4.** We chose lambda through an ablation on our validation set on Table 4.
>
> **5.** The interventional policy $\\pi(A|H)$ and how it differs from observation $P(A|H)$ is at the center of understanding the inverse propensity score under the potential outcome framework. $\\pi(A|H)$ is the ideal treatment assignment probability under an interventional (causal) environment while $P(A|H)$ is the probability of treatment assignment in observation, which has selection bias. Their ratio $w(H,A) = \\frac{\\pi(A|H)}{P(A|H)}$ is the inverse propensity score that we use to correct for the selection bias. This is a causal analog of importance sampling.
>
> **6.** The LSTM is used for the temporal propensity model, the generator is the noise generated by the conditional diffusion model. We will update Figure 1 to say "(Generator)" under where it says "Conditional Diffusion".
>
> **7.** This is a great point, we will add this sentence to Section 2.3 Advantages over VAEs and GANs: "Additionally, by fine-tuning from Stable Diffusion v1.5, we leverage pre-trained representations that understand natural image structure, enabling efficient training on limited medical data while operating in a compressed latent space that reduces computational requirements by approximately 8-fold compared to pixel-space generation."
>
> **Q1.** Yes, thank you we will fix that typo.
>
> **Q2.** Our dataset had a total of 4,505 patients with mean 5.66, median 6.0, and standard deviation 1.53 x-rays per patient, note we excluded patients with insufficient images to form earlier-later pairs. We will add this information to the dataset details section in Appendix A.1.
>
> **Q3.** The image shown in the historical patient covariates in the earlier image and the target image is later. There are big differences between the two, including lighting, camera angles, etc, which add to the complexity of the model task. This specific image pair is 48 months apart.
>
> **Q4.** This question raises an excellent point which are due to limitations of MSE, to alleviate this issue, we will also report R-squared between target and predicted image which is in the range of 0-1 so making it much more interpretable. Since the OAI dataset is the only known longitudinal interventional image dataset, we further justify the performance contribution by training on 5 different test splits which each hold out one of the 5 hospitals that are part of the OAI dataset. For example, TIDAL achieves an MSE of 0.136 ± 0.0004, a Gen MSE of 0.007 ± 0.0004, a Gen SSIM of 0.830 ± 0.0022, a KL Grade ITE error of 0.483 ± 0.0622, and a JSN Medial ITE error of 0.186 ± 0.0160.

---

### Meta-Review · Area_Chair_QCzy · 2026-01-06

**Summary:**

This paper develops a method for generating counterfactual medical images corresponding to treatments for patients with knee osteoarthritis.

Reviewer 8i99 appreciated the readability of the paper and the breadth of its contributions. As weaknesses, the reviewer pointed out the lack of synthetic data experiments, insufficiently comprehensive comparisons with existing methods, and questioned the algorithmic novelty. In response, the authors argued that synthetic experiments and broad comparisons were difficult because no existing methods are directly adapted to time-series settings, while nevertheless adding a small number of additional experiments. They also provided supplementary explanations regarding the claimed algorithmic novelty. However, at least from an external perspective, these rebuttals do not appear to resolve the reviewer’s concerns. Even if existing methods are not explicitly designed for time-series data, it is still possible—albeit potentially less effective—to apply them in a straightforward manner, and demonstrating this through comprehensive experiments is crucial. The added experiments are rather minimal and insufficient to address these concerns. Moreover, the added text intended to justify novelty remains abstract and focuses more on the novelty of the setting or data rather than clearly articulating a technical contribution.

Reviewer YNRX argued that the presentation was insufficient and that the work lacked technical novelty. The authors replied that the presentation would be improved in future revisions and offered several explanations in support of technical novelty, again emphasizing that existing methods cannot handle time-series data. However, as noted in the discussion of Reviewer 8i99’s comments, the authors did not succeed in clearly articulating genuine technical novelty, and it would be reasonable to view the approach as a straightforward application of existing methods. Furthermore, the claim that existing methods cannot handle time-series data is not sufficiently supported, either factually or experimentally, and thus fails to resolve the reviewer’s concerns.

Reviewer uVux evaluated the presentation and the contribution of the results positively, while raising concerns about potential instability arising from adversarial training and the need to validate inverse probability weighting (IPW). The authors demonstrated that they had conducted separate validations addressing each of these issues. These validations respond reasonably well to the specific questions raised by the reviewer.

Reviewer N3Su focused on the machine learning aspects of the paper and evaluated the experimental results as comprehensive. At the same time, the reviewer pointed out the absence of machine-learning-level novelty, the lack of theoretical guarantees, and insufficient evidence that the method can contribute broadly beyond the specific application domain. In response, the authors again provided explanations regarding technical novelty and argued that existing methods are not applicable to time-series settings. They did not offer a substantive response regarding theoretical insights. With respect to generalization, they mentioned the possibility of extending the approach to other diseases. While they explained the evaluation metrics used, they did not provide a convincing defense against concerns about their appropriateness. As with Reviewer 8i99’s comments, the authors did not succeed in clearly demonstrating technical novelty, and it is difficult to rule out the interpretation that the method is a simple application of existing techniques. The claim that prior methods cannot handle time-series data remains insufficiently supported by facts or experiments, and many of the reviewer’s concerns therefore remain unresolved. More broadly, several responses do not directly address the reviewers’ questions, leaving residual doubts.

In summary, while this work tackles an important problem and incorporates the meaningful idea of introducing temporal structure, it does not convincingly demonstrate that the proposed method is technically novel in a machine-learning sense. Comparisons with existing methods are limited and justified mainly by the claim that “existing methods are not specialized for time series,” which is insufficient to address the reviewers’ concerns. More generally, the reviewers’ focus appears to have been on the method as a general machine learning technique, rather than on domain-specific structures inherent to knee osteoarthritis or medical imaging. This suggests a mismatch between the paper’s content and its intended audience, raising the question of whether a machine learning conference such as ICLR is an appropriate venue. A venue more specialized in medical imaging may be better aligned with the paper’s goals and readership.

**Reviewer Concerns:**

See above.

**Reviewer Scores:**

See above

---

### Decision · Program_Chairs · 2026-01-26

Reject